# Restoration of Hepatic and Intestinal Integrity by *Phyllanthus amarus* Is Dependent on Bax/Caspase 3 Modulation in Intestinal Ischemia-/Reperfusion-Induced Injury

**DOI:** 10.3390/molecules27165073

**Published:** 2022-08-09

**Authors:** Ayobami Oladele Afolabi, Tunmise Maryanne Akhigbe, Adeyemi Fatai Odetayo, Davinson Chuka Anyogu, Moses Agbomhere Hamed, Roland Eghoghosoa Akhigbe

**Affiliations:** 1Department of Physiology, Ladoke Akintola University of Technology, Ogbomoso 210214, Oyo State, Nigeria; 2Department of Agronomy, Osun State University, Osogbo 210001, Osun State, Nigeria; 3Reproductive Biology and Toxicology Research Laboratory, Oasis of Grace Hospital, Osogbo 210001, Osun State, Nigeria or; 4Department of Physiology, University of Ilorin, Ilorin 240003, Kwara State, Nigeria; 5Department of Veterinary Pathology and Microbiology, University of Nigeria, Nsukka 410001, Enugu State, Nigeria; 6The Brainwill Laboratory, Osogbo 210001, Osun State, Nigeria; 7Department of Medical Laboratory Science, College of Medicine and Health Sciences, Afe Babalola University, Ado Ekiti 360001, Ekiti State, Nigeria

**Keywords:** *Phyllanthus amarus*, ischemia/reperfusion, oxidative stress, inflammation, apoptosis, hepatic function

## Abstract

Ethnopharmacological relevance: Oxidative stress is a key player in intestinal ischemia/reperfusion (I/R) injury (IIRI) with a tendency to trigger systemic inflammatory response, resulting in progressive distal organ injury. To date, the role of Bax/caspase 3 signaling in IIRI has not been reported. Furthermore, the discovery of a safe and effective drug remains pertinent in improving the outcome of IIRI. Therefore, this study investigated the role of Bax/caspase 3 signaling in intestinal I/R-induced intestinal and hepatic injury. In addition, the protective effect and possible associated mechanism of action of methanolic *Phyllanthus* *amarus* leaf extract (PA) against intestinal I/R-induced intestinal and hepatic injury were evaluated. Materials and methods: Fifty male Wistar rats were randomized into five groups (n = 10). The sham-operated group was received 0.5 mL of distilled water for seven days prior to the sham surgery, while the IIRI, febuxostat (FEB) + IIRI, low-dose PA (LDPA) + IIRI, and high-dose PA (HDPA) + IIRI groups underwent the I/R procedure. In addition to the procedure, IIRI, FEB + IIRI, LDPA + IIRI, and HDPA + IIRI received 0.5 mL of distilled water, 10 mg/kg of febuxostat, 200 mg/kg of PA, and 400 mg/kg of PA, respectively, for seven days prior to the I/R procedure. Results: Administration of methanolic *Phyllanthus* *amarus* leaf extracts attenuated the intestinal I/R-induced rise in intestinal and hepatic injury markers, malondialdehyde, nitric oxide, TNF-α, IL-6, and myeloperoxidase activities. In addition, *Phyllanthus* *amarus* ameliorated I/R-induced suppression of reduced glutathione, thiol and non-thiol proteins, and superoxide dismutase, catalase, and glutathione peroxidase activities in intestinal and hepatic tissues. These were coupled with the suppression of I/R-induced bacterial translocation, downregulation of I/R-induced activation of Bax/caspase 3 signaling, and improvement of I/R-induced distortion of intestinal and hepatic histoarchitecture by *Phyllanthus* *amarus*. Conclusion: Methanolic *Phyllanthus* *amarus* leaf extract protects against intestinal and hepatic injuries associated with intestinal I/R by suppressing oxidative-stress-mediated activation of Bax/caspase 3 signaling. The beneficial effects of *Phyllanthus* *amarus* may be ascribed to its constituent bioactive molecules, especially tannins, anthocyanin, alkaloids, and phenolics.

## 1. Introduction

Although virtually all organs are susceptible to ischemia reperfusion injury (IRI), intestinal ischemia–reperfusion injury (IIRI) has perhaps the greatest morbidity and mortality [1] among them all. This high morbidity and mortality are due to its propensity to affect virtually all organs, thus leading to multiple organ dysfunction syndrome (MODS) or, in extreme cases, multiple-organ failure (MOF) [2]. The mechanism by which this local event is transformed into life-threatening disorders has been the subject of numerous studies [3,4]. The pathophysiology of intestinal I/R-induced hepatic injury is complex and yet to be completely unraveled.

Restoration of blood flow results in biochemical and molecular metabolic alterations, which expose the tissues to damage mediated by free radicals [2] (generated by metabolic activities during the ischemic phase), leucocyte infiltration, inflammatory response, and apoptosis [5,6]. IIRI is especially lethal because of the disruption of the intestinal barrier, which results in increased permeability to toxins and bacteria, thereby allowing them to gain access to the general circulation [7]. The massive flux of intestinal microbial flora into portal circulation and, finally, into the systemic circulation [8,9] results in septicemia, which triggers a strong inflammatory response culminating in damage to distal organs such as the liver [10], lungs [11], kidney [12], heart [13], and even the brain [13]. Generally, the generation of reactive oxygen species plays a central role in the pathophysiological mechanism of IRI [5] through damage to cellular lipids [14], proteins [15], and DNA [16], which invariably facilitates apoptosis, eventually leading to cell death pathways [17]. Furthermore, ROS acting as a signaling molecule mediates the upregulation of pro-inflammatory cytokines, which disrupt intestinal barrier function [18], thereby promoting bacterial translocation. Although ROS-induced apoptosis has been shown to play a role in IIRI, there is a dearth of data reporting the role of Bcl-2-associated X protein (Bax) and cysteine-aspartic protease 3 (caspase 3) in I/R injury.

Caspase 3 may be activated via a death receptor (FAS, FAS-L), the endoplasmic reticulum, or the mitochondrial pathway (related to Bax) [19]. Upregulation of Bax triggers caspase 3 activation, which in turn induces apoptosis through chromosome condensation, cell membrane blebbing, and DNA fragmentation [20,21]. However, whether or not intestinal I/R-induced apoptosis involves Bax-mediated caspase 3 activation is not known.

Various molecules targeting ROS-mediated apoptosis have been employed in an attempt to protect against IRI. So far, no study has evaluated the role of *Phyllanthus amarus* (*P. amarus*) on intestinal I/R-induced injury. *P. amarus* has “slender, leaf-bearing branches, distichous leaves that are sub-sessile elliptic-oblong, rounded base and a branching annual glabrous herb that is about 30 to 60 cm high” [14] and belongs to the Euphorbiaceae family [15]. There are about 800 species of *P. amarus*, which are found in tropical and subtropical countries [16]. This herbal nutraceutical is known by various names. In Nigeria, it is called *Eyin Olobe* (Yoruba), *Gyeron tsuntsaye* (Hausa), and *ngwu* (Igbo). The leaves of *P. amarus* have been demonstrated to exert antioxidant, anti-inflammatory, analgesic, and antimicrobial properties [22,23]. It has also been reported to exhibit antidiabetic and antihypertensive activities [22] and to confer hepato-gastrointestinal protection [24,25].

Taking into consideration the medicinal values of *P. amarus* in folklore medicine, it is likely that *P. amarus* protects against I/R injury. Therefore, this study evaluated the role of Bax/caspase 3 signaling in intestinal I/R-induced intestinal and hepatic injury. In addition, the protective effect and the possible associated mechanism of action of methanolic *P. amarus* leaf extract against intestinal I/R-induced intestinal and hepatic injury was investigated.

## 2. Materials and Methods

### 2.1. Plant Collection

Fresh leaves of *Phyllanthus amarus* were collected in the California area, Ogbomoso, Oyo State, Nigeria. Identification and authentication of the plant were performed by Dr. Mrs. Ogundola of the Botany Unit, Department of Pure and Applied Biology, Ladoke Akintola University of Technology, Ogbomoso. The name of the plant was confirmed at http://www.theplantlist.org (accessed on 20 July 2021). A voucher specimen (LHO 615) was kept in the Herbarium at the Department of Botany, Ladoke Akintola University of Technology, Ogbomoso, Nigeria.

### 2.2. Preparation of Plant Extract

Preparation of plant extract has previously been reported [26,27]. The leaves of *Phyllanthus amarus* were air-dried for two weeks and pulverized using an electric blender. About 500 g of the obtained sample was soaked in 500 mL of 70% methanol for 72 h (3 days). The extract was filtered using mushin paper to separate the residue from the filtrate. The filtrate was poured inside a round-bottom flask of the Soxhlet apparatus and heated for one hour at a temperature of 60 °C, then poured inside a beaker and placed in a water bath for concentration at 100 °C for 24 h. The yield of the extract was 8.2%.

### 2.3. Phytochemical Analysis

Qualitative phytochemical analysis was performed using established methods described by Sofowora [28] and Trease et al. [29]. The secondary metabolites present were then quantified following established methods documented in previous studies [30,31,32,33,34].

### 2.4. Gas Chromatography–Mass Spectrophotometric Analysis

The bioactive components of methanolic *Phyllanthus amarus* leaf extract were identified by gas chromatography–mass spectrophotometric (GC-MS) (PelkinEler Inc., Waltham, MA, USA) analysis following established methods [35]. The database of the National Institute of Standard and Technology (NIST) was employed in interpreting the mass spectrum of GC-MS to ascertain the name, molecular weight, and structure of the bioactive components of methanolic *Phyllanthus amarus* leaf extract.

### 2.5. Experimental Animals

This study was carried out in the animal house of the Department of Physiology, Ladoke Akintola University of Technology, Ogbomoso, Nigeria. Ethical approval was issued by The Ministry of Health, Oyo State, Nigeria (Approval Number AD13/479/44406). Fifty adult male Wistar rats of similar weight (190 ± 5 g) were used for the current study. The animal model using Wistar rats was adopted to mimic the clinical presentation of IIRI because this strain of rats has been established to share a close relationship to humans per genetics and biological and behavioral characteristics [36]. Animals were allowed free access to standard rat chow and water. Animals were humanely cared for using the Guidelines for the Care and Use of Laboratory Animals as published by the U.S. National Institutes of Health (NIH Publication No. 85-23, revised 1996).

### 2.6. Experimental Design

Animals were acclimatized for two weeks, then randomly assigned to five groups, each consisting of ten rats. The sham group was sham-operated and received 0.5 mL of distilled water for seven days prior to the sham surgery, while the IIRI, febuxostat (FEB) + IIRI, low-dose methanolic *Phyllanthus amarus* leaf extract (LDPA) + IIRI, and high-dose methanolic *Phyllanthus amarus* leaf extract (HDPA) + IIRI groups underwent the I/R procedure. In addition to the procedure, IIRI, FEB + IIRI, LDPA + IIRI, and HDPA + IIRI received 0.5 mL of distilled water, 10 mg/kg of febuxostat, 200 mg/kg of methanolic *Phyllanthus amarus* leaf extract, and 400 mg/kg of PA, respectively, for seven days prior to the I/R procedure. Febuxostat was used as a standard control drug with anti-inflammatory and antioxidant properties that protect against I/R injury [37,38]. The administration of the drugs was via gavage. The doses of febuxostat [39] and *Phyllanthus amarus* [40,41] were as previously reported.

IIRI was induced as previously reported [42] with some modifications. The rats were weighed and anesthetized with 10 mg/kg of Xylazine and 50 mg/kg of Ketamine. The abdomen was cleaned with 10% povidone iodine; a ventral midline incision was made on the abdomen, and the intestine was mobilized. About 5 cm of the ileum was measured proximally from the illio-cecal junction and a further 5 cm measured proximal to the first 5 cm. The most proximal five cm of ileum was folded on itself and twisted 720° clockwise. The twisted ileum was then anchored to the anterior abdominal wall by placing a 3-0 chromic suture around it and through the avascular part of the mesentery. The abdomen was then closed lightly with a 3-0 chromic suture, after which the animal was left for forty-five minutes. At the expiration of the 45 min of ischemia, the abdomen was reopened. Loss of palpitation was observed to confirm that ischemia had been successfully induced. Reperfusion was induced by removing the suture and untwisting the ileum for twenty-four hours.

### 2.7. Sacrifice and Tissue Collection

After 60 min of reperfusion, animals were euthanized and the intestine and liver were rapidly removed and weighed after separating the adhering structures.

Portions of the intestinal and hepatic tissues were homogenized in an appropriate volume of cold phosphate-buffered saline using a glass homogenizer. The homogenates were centrifuged at 10,000 rpm for 15 min in a cold centrifuge to obtain the supernatant fractions.

In addition, portions of the intestinal and hepatic tissues were obtained and fixed in 10% formalin-phosphate-buffered saline at 4 °C overnight. The samples were dehydrated and embedded in paraffin wax. About 5 µm-thick sections were cut and stained with hematoxylin-eosin and examined under a light microscope by a pathologist who was blinded to the study protocol. Photomicrographs were taken at ×100 and ×400 magnifications.

Formalin-fixed and paraffin-embedded intestinal tissues were sectioned at about 4 µm for immunohistochemistry for Bax protein expression, as earlier reported [43].

### 2.8. Biochemical Analyses

#### 2.8.1. Tissue Injury Markers

The activities of aspartate transaminase (AST), alanine transferase (ALT), alanine phosphatase (ALP), and gamma-glutamyl transferase (GGT) in the hepatic tissue [44], as well as the activities of lactate dehydrogenase (LDH) in the intestinal and hepatic tissues were assayed by spectrophotometry, as previously reported [21].

#### 2.8.2. Markers of Oxidative Stress and Antioxidant Levels

The intestinal and hepatic concentrations of malondialdehyde (MDA) and reduced glutathione (GSH) [5] and the activities of superoxide dismutase (SOD), catalase, and glutathione peroxidase (GPx) [17,45] in the intestinal and hepatic tissues were assayed by spectrophotometry, as previously reported.

Thiol and non-thiol proteins were assayed by colorimetry, as earlier reported [46,47].

#### 2.8.3. Markers of Inflammation

The concentrations of nitric oxide (NO) and myeloperoxidase (MPO) activities in the intestinal and hepatic tissues were determined by colorimetric methods as earlier documented [17]. The intestinal and hepatic levels of tumor necrotic factor-α (TNF-α) and interleukin-6 (IL-6) were determined using ELISA kits (Elabscience Biotechnology Inc., Houston, TX, USA) according to the manufacturers’ guidelines.

#### 2.8.4. Marker of Apoptosis

The activities of caspase 3 in the intestinal and hepatic tissues were assayed using ELISA kits (Elabscience Biotechnology Inc., Houston, TX, USA) according to the manufacturer’s guidelines.

### 2.9. Microbiological Analysis

Bacterial translocation was determined as previously reported [48] under strict sterile conditions. Briefly, some portions of the intestinal and hepatic tissues were cut into pieces using a sterile blade and put into 1 mL of Mueller–Hinton Broth. The samples were homogenized, and about 100 µL of each sample was inoculated into nutrient agar (NA), eosin-methylene blue (EMB), and Mac Conkey agar (MCCA) for colony counts. The cultures were incubated for 48 h and observed for the presence of growth under either aerobic or anaerobic conditions.

### 2.10. Histopathological Analysis

Intestinal and hepatic histomorphological appearance was scored using Chiu’s [49] (Appendix A) and Eckhoff’s score [50] (Appendix A), respectively.

Digital photomicrographs obtained from histopathology and immunohistochemistry were imported into the Image J software (NIH, Bethesda, MD, USA) with specific plugins for quantification of intestinal villi length and crypt depth, hepatic sinusoidal length and width, and Bax expression. The results were expressed as fold change relative to the normal control group [43].

### 2.11. Statistical Analysis

GraphPad Prism (Versions 7.00; Graph Pad Software, San Diego, CA 92108, USA) was used for data analysis. Analysis of variance (ANOVA) was used to compare the mean values across all the groups, and then, Tukey’s post hoc test was used for pairwise comparison. To ascertain intestinal–hepatic crosstalk, Pearson’s correlation was conducted between hepatic injury markers and intestinal histomorphological changes (Chiu’s score). Data are presented as means ± standard deviations.

## 3. Results

### 3.1. Bioactive Compounds of Methanolic Phyllanthus amarus Leaf Extract

Phytochemical analysis revealed that tannins (1.850 ± 0.030) and anthocyanin (1.820 ± 0.020) were the most abundant phytochemical compounds in methanolic *P. amarus* leaf extract, followed by alkaloids (1.607 ± 0.013), phenolics (1.130 ± 0.014), glycosides (0.863 ± 0.015), saponins (0.747 ± 0.017), and triterpenoids (0.140 ± 0.010) (Table 1).

Table 2 and Figure 1 show the organic constituents of methanolic *P. amarus* as revealed by GC-MS. It was observed that methyl stearate had the highest retention time of 16.655 min, while butanoic acid was found to have the least retention time of 3.574 min. The chemical properties of these organic constituents are shown in Table 2, and the chemical structures are provided in Appendix A.

### 3.2. Organ Weight

As shown in Figure 2, I/R led to a significant increase in the absolute intestinal and hepatic weight compared with the animals in the sham group. The I/R-induced increase in intestinal and hepatic mass was significantly attenuated by FEB, LDPA, and HDPA.

### 3.3. Hepatic Injury Markers

The effect of intestinal ischemia/reperfusion injury on tissue injury markers was also probed (Table 3). It was observed that intestinal I/R significantly elevated the activities of hepatic AST, ALT, ALP, and GGT compared with the sham group. The observed changes in hepatic injury markers following I/R were significantly ameliorated by FEB, LDPA, and HDPA. Although the effects of HDPA on ALT, ALP, and GGT were comparable with those of FEB, HDPA significantly reduced AST activity when compared with FEB.

In addition, I/R led to a marked increase in the activities of intestinal and hepatic LDH, as shown in Figure 3. The observed rise in intestinal and hepatic LDH was blunted by FEB, LDPA, and HDPA. When compared with FEB, HDPA caused a significant reduction in intestinal LDH activity.

### 3.4. Markers of Oxidative Stress and Inflammation

I/R led to a significant increase in intestinal and hepatic MDA. This increase was significantly attenuated by FEB, LDPA, and HDPA. Furthermore, I/R led to a significant reduction in intestinal and hepatic GSH, thiol, and non-thiol proteins, as well as SOD, catalase, and GPx activities. The observed decrease in these parameters was abolished by FEB, LDPA, and HDPA treatments. When compared with other treatments, HDPA caused a significantly reduced intestinal MDA level and increased GSH and non-thiol protein levels and SOD activities (Table 4).

Furthermore, intestinal I/R led to a significant increase in intestinal and hepatic MPO activities and NO, TNF-α, and IL-6 levels compared with the sham-operated group. The I/R-led increase in these inflammatory markers was significantly prevented by the FEB, LDPA, and HDPA treatments. When compared with other treatments, HDPA caused significantly reduced intestinal and hepatic MPO activities and hepatic NO and TNF-α levels (Figure 4).

### 3.5. Intestinal and Hepatic Histoarchitecture

As shown in Figure 5, intestinal I/R disrupted intestinal histoarchitecture. The sham-operated animals showed normal villi from the mucosal layer, and the lumen was moderately infiltrated with lymphocytes. The propria and submucosal layer showed moderate infiltration of inflammatory cells. The animals in the IIRI group exhibited a chronically inflamed mucosal layer with sloughed and inflamed villi. There was infiltration of the lumen by lymphocytes, polymorphs, and plasma cells involving the propria and glands. The submucosal layer was infiltrated severely by inflammatory cells. Animals who received the FEB treatment showed moderately preserved villi from the mucosal layer. The lumen showed severe lymphocyte infiltration and the propria with severe inter-glandular infiltration of inflammatory cells. The submucosal layer appeared normal. Unlike the animals in the LDPA-treated group, which showed moderately preserved villi from the mucosal layer, those in the HDPA-treated group showed normal villi from the mucosal layer. In addition, the animals in both the LDPA- and HDPA-treated groups showed lumen and propria with mild lymphocyte infiltrations and inter-glandular infiltration of inflammatory cells, respectively. The submucosal layer appeared normal in the LDPA- and HDPA-treated animals.

As shown in Figure 6, the hepatic histoarchitecture was distorted following intestinal I/R compared with the animals in the sham group. Animals in the IIRI group showed mild congestion of the portal vein, some degenerated hepatocytes, and necrotic hepatocytes. At the same time, all these histopathological lesions were prevented by the FEB, LDPA, and HDPA treatments.

Figure 7 shows the histomorphological changes in the intestinal and hepatic tissues using Chiu’s and Eckhoff’s scores, respectively. When compared with the sham-operated rats, I/R caused a significant increase in Chiu’s and Eckhoff’s scores in the IIRI group. Treatments with FEB, LDPA, and HDPA significantly abolished I/R-induced alterations in intestinal and hepatic histomorphology. IIRI also significantly reduced villi length and crypt depth compared with the sham-operated rats. The observed IIRI-induced alterations in villi length and crypt depth were significantly ameliorated by FEB, LDPA, and HDPA treatments. Although the effects of LDPA and HDPA were similar on crypt depth, the effect on villi length was dose-dependent. Additionally, IIRI significantly reduced hepatic sinusoidal length and width. The noted IIRI-led reduction in sinusoidal length, but not sinusoidal width, was significantly alleviated by the FEB, LDPA, and HDPA treatments. Only FEB and HDPA significantly abrogated IIRI-driven sinusoidal width reduction.

### 3.6. Apoptotic Markers

Animals in the IIRI group showed a marked rise in intestinal and hepatic caspase 3 activities compared with the sham group. The I/R-driven rise in caspase 3 activities was significantly alleviated by the FEB, LDPA, and HDPA treatments (Figure 8).

### 3.7. Bax Expression

When compared with the sham-operated group, the IIRI group showed a marked increase in Bax expression in the intestinal tissues. Treatments with FEB, LDPA, and HDPA significantly inhibited I/R-induced upregulation of Bax expression. Additionally, HDPA treatment significantly reduced intestinal Bax expression when compared with other treatments (Figure 9).

### 3.8. Bacterial Translocation

Using nutrient agar, eosin-methylene blue, and Mac Conkey agar as culture media, I/R significantly increased the bacteria count in intestinal and hepatic tissues in the IIRI group compared with the sham-operated group. FEB treatment significantly reduced the I/R-led rise in bacteria count. Interestingly, when compared with the IIRI and FEB + IIRI groups, the LDPA and HDPA groups showed significantly reduced bacteria counts (Figure 10).

IR promotes reactive oxygen species (ROS) and pro-inflammatory cytokine generation, which in turn translocate Bax from the cytosol into the mitochondrial outer membrane, where it becomes activated by binding with BH3-only proteins. Activated Bax oligomerizes to form pores in the mitochondrial outer membrane and triggers the release of cytochrome c into the cytosol, which in turn activates caspase 3, which induces apoptosis. Administration of PA inhibits IR-induced upregulation of Bax/caspase-3-mediated apoptosis via suppression of ROS and cytokine generation.

### 3.9. Correlation Study

Using Pearson’s correlation analysis, it was observed that Chiu’s score of intestinal histomorphological changes was significantly positively correlated with hepatic Eckhoff’s histomorphological score, AST, and ALT, but not GGT. Furthermore, intestinal damage (using Chiu’s score) accounted for 73.12% of hepatic Eckhoff’s score, 87.04% of AST, and 77.84% of ALT (Table 5).

## 4. Discussion

IIR is a surgical emergency with high morbidity and mortality [51]. Although the pathological events involved in I/R are complex, the primary cause of organ injury has been established to be ischemia, which is superimposed on secondary tissue damage induced by reperfusion. The crosstalk between the intestine and other organs increases the susceptibility of vital organs to I/R injury in IIRI. Hence, the development of a safe and effective management of IIRI is pertinent in improving patients’ outcomes [52]. In the current study, we provided the seemingly first evidence that Bax/caspase 3 pathway activation can result in IIRI and hepatic injury via an oxidative-stress-sensitive signaling. Our present findings also shine a light on the protective mechanism of methanolic *P. amarus* leaf extract in I/R-induced intestinal–hepatic damage.

Bax proteins are nuclear-encoded proteins that are present in higher eukaryotes and are capable of piercing the outer membrane of the mitochondria to induce cell death by apoptosis [53]. Bax adopts two stable conformational states: the native Bax, which is in the cytosol, and the fully activated Bax, which is in the mitochondria [54]. In a healthy state, Bax is primarily a cytosolic protein with just a fraction in the mitochondria [55,56] and endoplasmic reticulum [57,58] via constant retranslocation from the mitochondria to the cytosol. However, during cellular stress, induced by oxidative stress or any form of stress, Bax is translocated from the cytosol to insert into the mitochondrial outer membrane and become activated by binding with BH3-only proteins [53]. BH3-only protein may also bind Bcl-2-like proteins to indirectly activate Bax [53]. Once activated, Bax oligomerizes to form pores in the mitochondrial outer membrane to trigger the release of cytochrome c into the cytosol and activates caspase 3 with consequent apoptosis [53]. Thus, IIRI-driven upregulation of caspase 3 activity observed in this study may be a consequence of IIRI-led upregulation of Bax expression. It is noteworthy that methanolic *P. amarus* leaf extract inhibited IIRI-led upregulation of Bax/caspase 3 signaling. Although there is no available study in the literature that documents the effect of *P. amarus* on IIRI, the present findings provide an extension of the reports of Guha and his colleagues [59] on the anti-apoptotic effect of *P. amarus*. It is likely that this botanical herb inhibits the translocation of Bax into the mitochondrial outer membrane and suppresses cytochrome c release into the cytosol, thus stultifying Bax/caspase-3-mediated apoptosis [59].

The observed I/R-induced oxidative stress evidenced by elevated intestinal and hepatic MDA concentrations and reduced GSH and thiol and non-thiol protein levels that are accompanied by the suppressed activities of enzymatic antioxidants is a possible trigger for I/R-driven activation of Bax/caspase 3 signaling [53]. The observed oxidative stress following I/R is in consonance with previous studies [5,51,60]. The observed elevated MDA is a reflection of I/R-led accumulation of reactive oxygen species (ROS), especially superoxide and hydrogen peroxide. Superoxide radicals are rapidly dismutated to hydrogen peroxide [61], which readily diffuses through the mitochondrial membrane to elicit lipid peroxidation, protein denaturation, and oxidative damage to the nucleic acids [62]. Hence, the activity of methanolic *P. amarus* leaf extract against MDA accumulation may be a likely mechanism for alleviating I/R-induced Bax/caspase 3 activation.

It is an established fact that the generated free radicals are decimated by the glutathione system, catalase, and the non-thiol and thiol proteins, including peroxiredoxins and GPx [63], which makes them the target for ROS. Available data in the literature have shown that upregulation of these enzymatic and non-enzymatic antioxidants protect cells from ROS-induced apoptosis [5,45,64]. The findings of this study revealed that the glutathione system and catalase, as well as the thiol and non-thiol proteins were depleted in the intestinal and hepatic tissues following IIRI, indicating a consumptive effect of I/R-led ROS generation on the antioxidant system.

Furthermore, I/R-induced intestinal and hepatic damage evidenced by distorted histoarchitecture of the intestine and liver, as well as elevated injury markers (AST, ALT, ALP, GGT, and LDH) may be due to ROS-led enhancement of MPO activity, which promotes infiltration of polymorphonuclear leucocytes, increased NO generation, and accumulation of pro-inflammatory cytokines (TNF-α and IL-6) [21]. NO plays a central role in the maintenance of tissue integrity [65]. Although endothelial nitric-oxide-synthase (eNOS)-induced NO is protective at the onset of injury, it is dysfunctional during oxidative stress [66]. Superoxide radicals react with NO to generate peroxynitrite, which in turn elicits oxido-inflammatory damage [67]. Pro-inflammatory cytokines have also been shown to trigger caspase-3-dependent apoptosis directly or indirectly by promoting ROS generation [61]. The upregulation of MDA and inflammatory markers and downregulation of antioxidants in the intestinal and hepatic tissues confirm the synchronous intestinal–hepatic damage in intestinal I/R-induced injury. The positive correlation between hepatic injury markers and the distortion of intestinal histomorphology also underscores the intestinal–hepatic crosstalk in intestinal I/R injury. The rescue effect of methanolic *P. amarus* leaf extract against I/R-induced oxidative stress and inflammation may be ascribed to the effective upregulation of the antioxidant system, suppression of MPO activity, and accumulation of NO and pro-inflammatory cytokines by the herbal nutraceutical.

Among all organs and tissues that are susceptible to IRI, the intestines are unique in being capable of bacterial translocation [48,68]. This unique ability may be a major contributor to the high mortality of IIRI. The *P. amarus* extract reduced bacterial translocation to the liver in a dose-dependent manner, with the high dose almost totally preventing translocation into the liver. Besides its antioxidant, anti-inflammatory, and antiapoptotic functions, the well-documented antibacterial activity could have contributed greatly to the near total stultification of bacterial translocation observed in this study.

The biological activities of methanolic *P. amarus* leaf extract against I/R-induced intestinal/hepatic damage may be attributed to its bioactive constituent compounds. Tannins, the most abundant phytochemicals in the botanical, have been reported to possess antioxidant and anti-apoptotic properties, as well as antimicrobial activities via the reduction of cytochrome c release [69,70,71]. Anthocyanins have also been shown to exert antioxidant and anti-inflammatory effect by scavenging superoxide radicals and peroxynitrites, as well as inducing enzymatic antioxidants [72,73]. The antioxidant and antibacterial activities of alkaloids and phenols have also been well documented [74,75,76,77]. Additionally, butanoic acid is a short-chain fatty acid that serves as an energy substrate and regulator of colonocytes’ reproduction [78]. It exerts anti-inflammatory and antibacterial activities by downregulating the expression of pro-inflammatory cytokines, adhesion molecules, and nuclear factor kappa-B (NF-kB), as well as the lipopolysaccharide-induced rise in IL-8 production [78]. Methenamine has also been established to possess antibacterial activities and has been demonstrated to be useful in multi-drug resistance [79]. Recent evaluations of biological activity in a series of thiazole-based agents showed that the introduction of a cyclobutane fragment into the thiazole ring led to significantly improved antifungal and antibacterial activities in the compound [80]. Octanoids and oleic acids have been demonstrated to exert anti-inflammatory activity by inhibiting NF-kB signaling and upregulating peroxisome-proliferator-activated receptors (PPARγ) [81,82,83,84]. Citric acid has also been shown to be a potent antioxidant and anti-inflammatory molecule by its ability to enhance the glutathione system and inhibit eNOS expression and accumulation of cytokines [85]. Acetic acid has been revealed to dampen inflammation and exert a bactericidal effect [86]. Furthermore, methyl stearate has been shown to possess antioxidant properties [87].

## 5. Conclusions

This study showed that intestinal I/R injury is mediated by the activation of Bax/caspase 3 signaling. Administration of methanolic *P. amarus* leaf extract could prevent intestinal and hepatic injuries associated with intestinal I/R by suppressing oxidative stress-mediated activation of Bax/caspase 3 signaling (Figure 11). This provides a novel mechanistic understanding of the pathogenesis of intestinal–hepatic crosstalk in intestinal I/R and would be beneficial in the design of a specific inhibitor of Bax/caspase 3 signaling that is safe and effective.

## Figures and Tables

**Figure 1 molecules-27-05073-f001:**
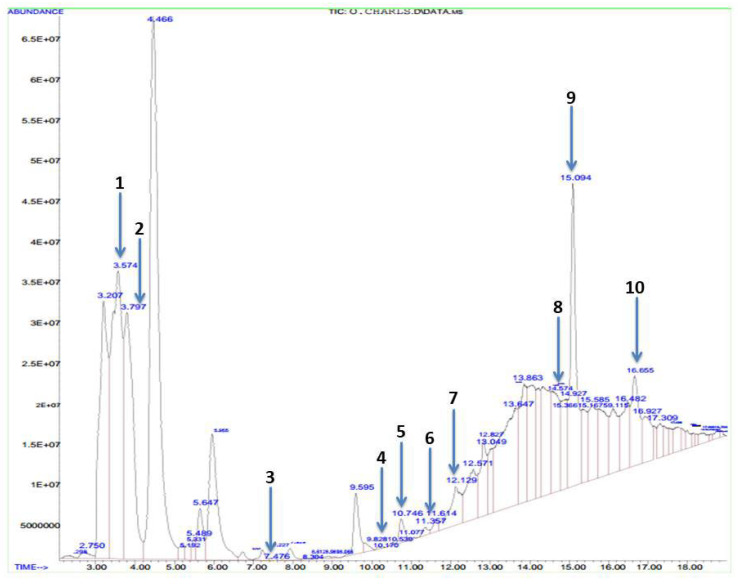
GC-MS analysis of methanolic *P. amarus* leaf extract. 1: Butanoic acid, 2: methenamine, 3: cyclobutane, 4: octanoic acid, 5: thiazole acid, 6: oleic acid, 7: citric acid, 8: acetic acid, 9: hexadecanoic acid, 10: methyl ester.

**Figure 2 molecules-27-05073-f002:**
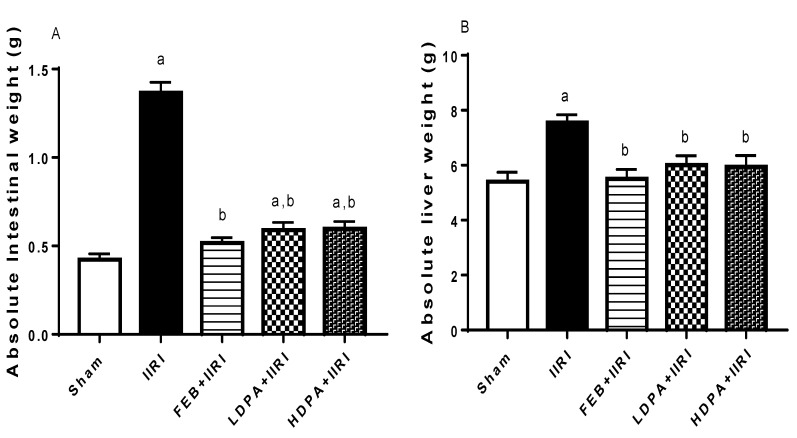
Effect of intestinal ischemia/reperfusion (I/R) and methanolic *Phyllanthus amarus* leaf extract on intestinal and hepatic weight. FEB, LDPA, and HDPA significantly reduced IIRI-induced increase in absolute intestinal weight (**A**) and absolute hepatic weight (**B**). IIRI: intestinal ischemia/reperfusion injury, FEB: febuxostat, LDPA: low-dose *Phyllanthus amarus*, HDPA: high-dose *Phyllanthus amarus*, ^a^
*p* < 0.05 versus sham, ^b^ *p* < 0.05 versus IIRI. Values represent the mean for ten replicates ± the standard deviation.

**Figure 3 molecules-27-05073-f003:**
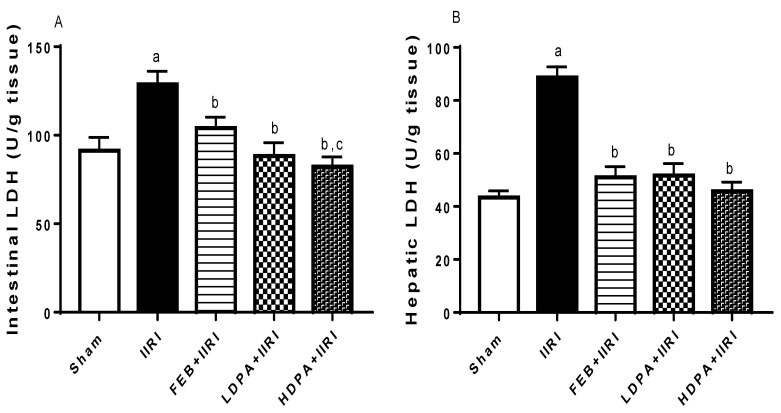
Effect of intestinal ischemia/reperfusion (I/R) and methanolic *Phyllanthus amarus* leaf extract on intestinal and hepatic lactate dehydrogenase (LDH) activities. FEB, LDPA, and HDPA significantly reduced IIRI-induced rise in intestinal LDH (**A**) and hepatic LDH (**B**). IIRI: intestinal ischemia/reperfusion injury, FEB: febuxostat, LDPA: low-dose *Phyllanthus amarus*, HDPA: high-dose *Phyllanthus amarus*, ^a^ *p* < 0.05 versus sham, ^b^ *p* < 0.05 versus IIRI, ^c^ *p* < 0.05 versus FEB + IIRI. Values represent the mean for ten replicates ± the standard deviation.

**Figure 4 molecules-27-05073-f004:**
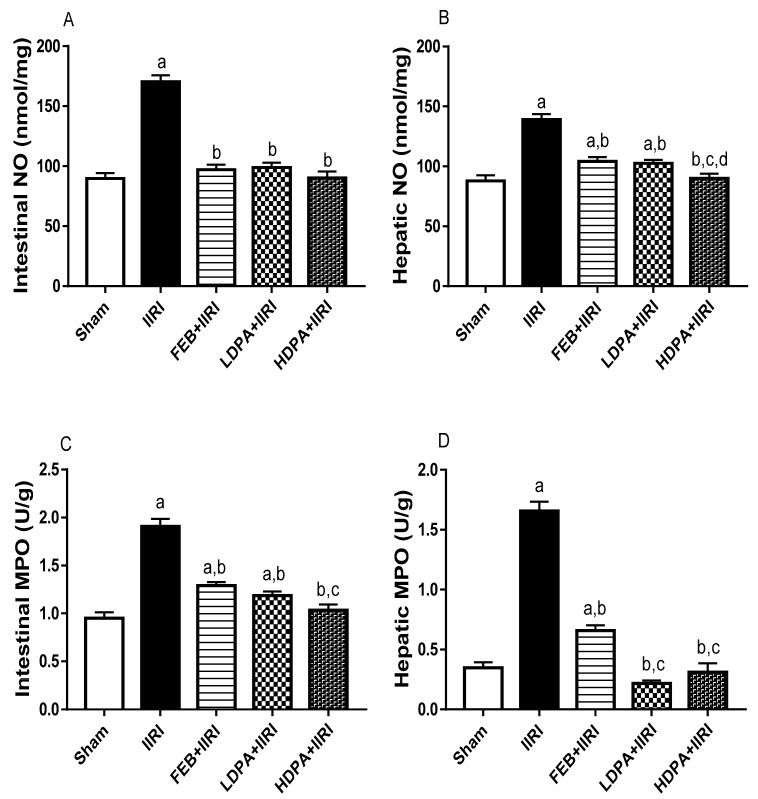
Effect of intestinal ischemia/reperfusion (I/R) and methanolic *Phyllanthus amarus* leaf extract on markers of inflammation in the intestinal and hepatic tissues. FEB, LDPA, and HDPA significantly reduced IIRI-induced increase in intestinal NO (**A**) and hepatic NO (**B**), intestinal MPO (**C**) and hepatic MPO activity (**D**), intestinal TNF-α (**E**), hepatic TNF-α (**F**), intestinal IL-6 (**G**) and hepatic IL-6 (**H**) levels. IIRI: intestinal ischemia/reperfusion injury, FEB: febuxostat, LDPA: low-dose *Phyllanthus amarus*, HDPA: high-dose *Phyllanthus amarus*, ^a^ *p* < 0.05 versus sham, ^b^ *p* < 0.05 versus IIRI, ^c^ *p* < 0.05 versus FEB + IIRI, ^d^ *p* < 0.05 versus LDPA + IIRI. Values represent the mean for ten replicates ± the standard deviation.

**Figure 5 molecules-27-05073-f005:**
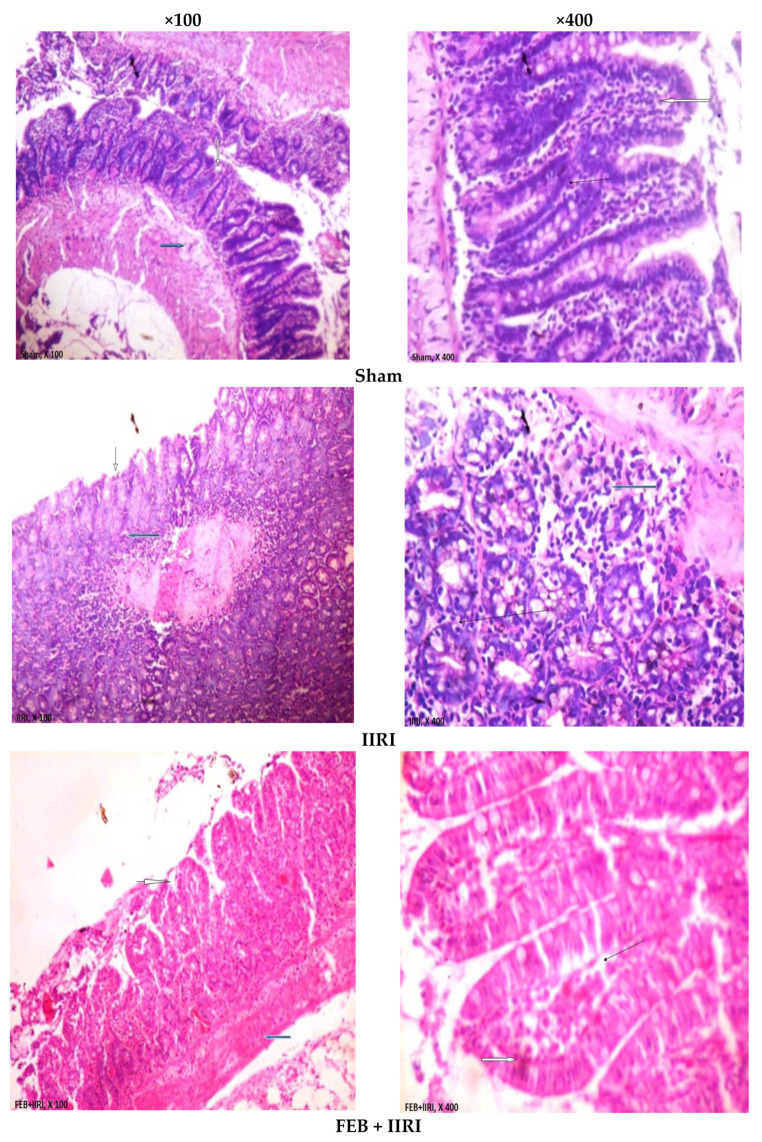
Effect of intestinal ischemia/reperfusion (I/R) and methanolic *Phyllanthus amarus* leaf extract on intestinal histoarchitecture. IIRI: intestinal ischemia/reperfusion injury, FEB: febuxostat, LDPA: low-dose *Phyllanthus amarus*, HDPA: high-dose *Phyllanthus amarus.* The sham-operated animals showed normal villi from the mucosal layer (white arrow). The lumen showed moderate lymphocyte infiltration, while the propria showed moderate inter-glandular infiltration of inflammatory cells (slender arrow), and the submucosal layer was moderately infiltrated by inflammatory cells (blue arrow). Animals in the IIRI group showed a chronically inflamed mucosal layer with sloughed and inflamed villi (white arrow). The lumen showed infiltration of lymphocytes, polymorphs, and plasma cells involving the propria and glands (slender arrow), while the submucosal layer was severely infiltrated by inflammatory cells (blue arrow). Rats in the FEB + IIRI group showed moderately preserved villi from the mucosal layer (white arrow). The lumen showed severe lymphocyte infiltration; the propria showed severe inter-glandular infiltration of inflammatory cells (slender arrow); the submucosal layer appeared normal (blue arrow). Rats in the LDPA + IIRI showed moderately preserved villi from the mucosal layer (white arrow). The lumen showed mild lymphocyte infiltration; the propria showed mild inter-glandular infiltration of inflammatory cells (slender arrow); the submucosal layer appeared normal (blue arrow). The animals in the HDPA + IIRI showed normal villi from the mucosal layer (white arrow). The lumen showed mild lymphocyte infiltration; the propria showed mild inter-glandular infiltration of inflammatory cells (slender arrow); the submucosal layer appeared normal (blue arrow).

**Figure 6 molecules-27-05073-f006:**
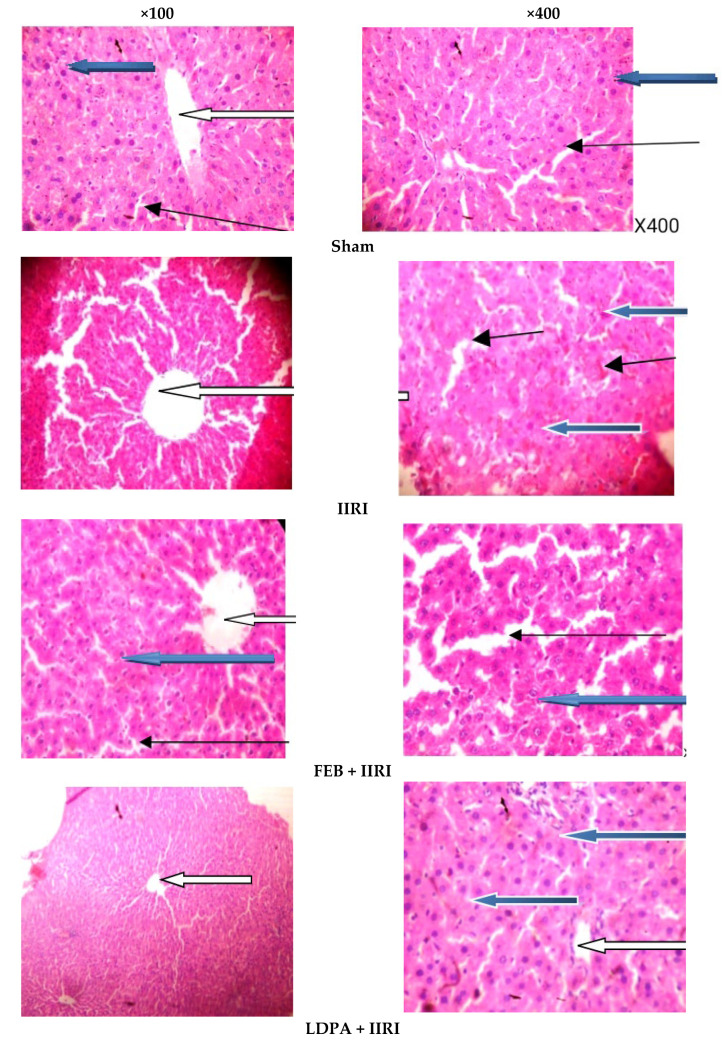
Effect of intestinal ischemia/reperfusion (I/R) and methanolic *Phyllanthus amarus* leaf extract on hepatic histoarchitecture. IIRI: intestinal ischemia/reperfusion injury, FEB: febuxostat, LDPA: low-dose *Phyllanthus amarus*, HDPA: high-dose *Phyllanthus amarus.* The sham-operated animals showed normal central venules (white arrow) and normal hepatocytes (blue arrow), and the sinusoids also appeared normal (slender arrow). No pathological lesions were seen. Animals in the IIRI group showed normal central venules (white arrow), mild congestion of the portal vein (black arrow), some degenerated hepatocytes (blue arrow), and some necrotic hepatocytes (red row). The sinusoids appeared normal (slender arrow). The rats in the FEB + IIRI group showed normal central venules without congestion (white arrow). The hepatocytes appeared normal (blue arrow), and the sinusoids appeared normal without being infiltrated by inflammatory cells (slender arrow). Animals in the LDPA + IIRI group showed normal central venules (white arrow), and the hepatocytes (blue arrow) and sinusoids (slender arrow) appeared normal. The rats in the HDPA + IIRI group showed normal central venules (white arrow). The hepatocytes appeared normal (blue arrow), and the sinusoids also appeared normal and not infiltrated (slender arrow).

**Figure 7 molecules-27-05073-f007:**
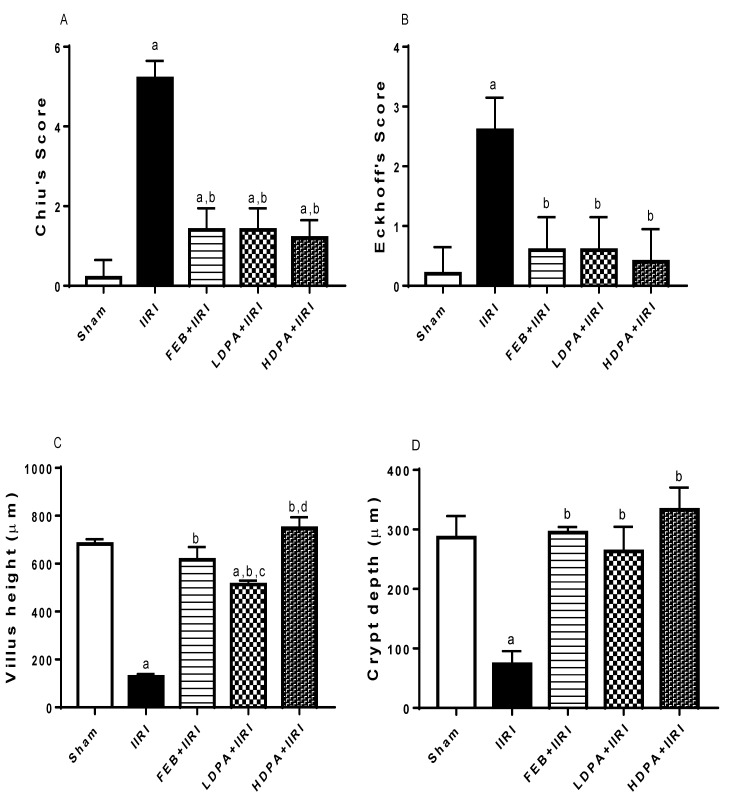
Effect of intestinal ischemia/reperfusion (I/R) and methanolic *Phyllanthus amarus* leaf extract on intestinal and hepatic histopathological changes using Chiu’s and Eckhoff’s score, respectively, intestinal villi height and crypt depth, and hepatic sinusoidal length and width. FEB, LDPA, and HDPA significantly reduced IIRI-induced rise in Chiu’s score (**A**) and Eckhoff’s score (**B**), as well as increase increased IIRI-induced reduction in villus height (**C**), crypt depth (**D**), sinusoidal length (**E**) and sinusoidal width (**F**). IIRI: intestinal ischemia/reperfusion injury, FEB: febuxostat, LDPA: low-dose *Phyllanthus amarus*, HDPA: high-dose *Phyllanthus amarus*, ^a^ *p* < 0.05 versus sham, ^b^ *p* < 0.05 versus IIRI, ^c^ *p* < 0.05 versus FEB+IIRI, ^d^ *p* < 0.05 versus LDPA+IIRI. Values represent the mean for five replicates ± the standard deviation.

**Figure 8 molecules-27-05073-f008:**
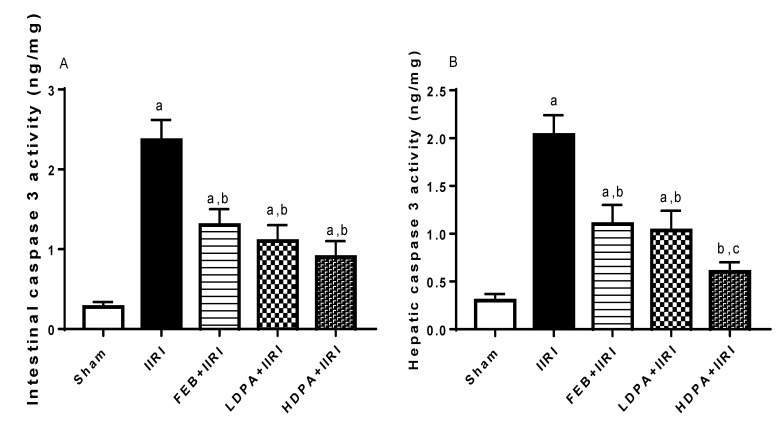
Effect of intestinal ischemia/reperfusion (I/R) and methanolic *Phyllanthus amarus* leaf extract on intestinal and hepatic caspase 3 activities. FEB, LDPA, and HDPA significantly reduced IIRI-induced rise in intestinal (**A**) and hepatic (**B**) caspase 3 activities. IIRI: intestinal ischemia/reperfusion injury, FEB: febuxostat, LDPA: low-dose *Phyllanthus amarus*, HDPA: high-dose *Phyllanthus amarus*, ^a^ *p* < 0.05 versus sham, ^b^ *p* < 0.05 versus IIRI, ^c^ *p* < 0.05 versus FEB + IIRI. Values represent the mean for ten replicates ± the standard deviation.

**Figure 9 molecules-27-05073-f009:**
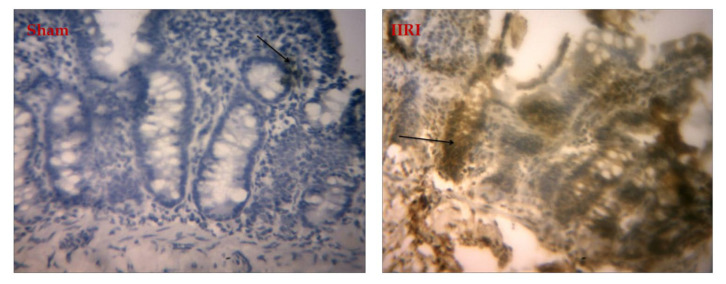
Effect of intestinal ischemia/reperfusion (I/R) and methanolic *Phyllanthus amarus* leaf extract on intestinal Bax expression. IIRI: intestinal ischemia/reperfusion injury, FEB: febuxostat, LDPA: low-dose *Phyllanthus amarus*, HDPA: high-dose *Phyllanthus amarus*, ^a^ *p* < 0.05 versus sham, ^b^ *p* < 0.05 versus IIRI, ^c^ *p* < 0.05 versus FEB + IIRI, ^d^ *p* < 0.05 versus LDPA + IIRI. Values represent the mean for five replicates ± the standard deviation.

**Figure 10 molecules-27-05073-f010:**
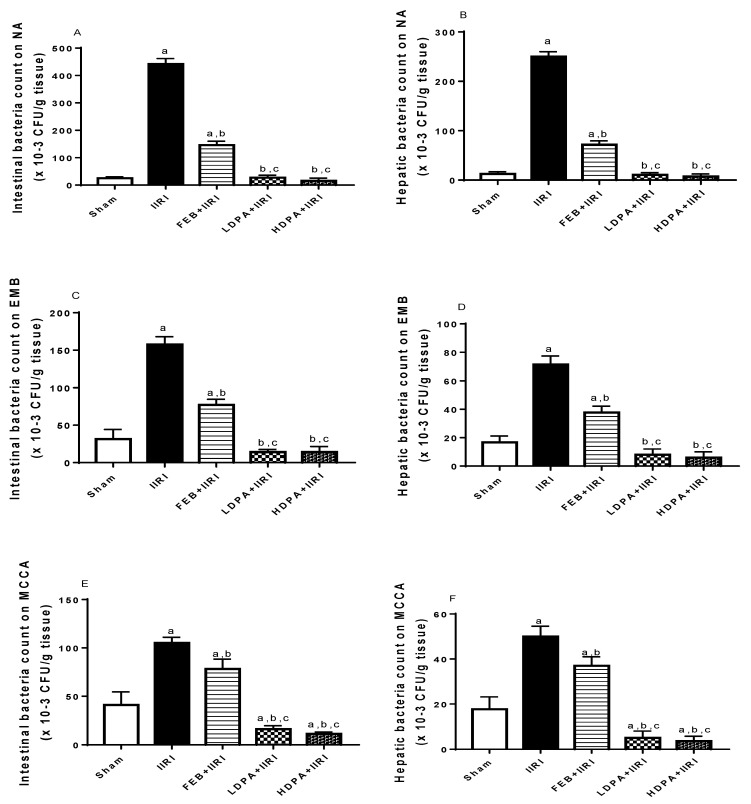
Effect of intestinal ischemia/reperfusion (I/R) and methanolic *Phyllanthus amarus* leaf extract on bacteria translocation. FEB, LDPA, and HDPA significantly reduced IIRI-induced increase in intestinal and hepatic bacteria count using nutrient agar (NA), (**A**,**B**), Eosin-Methylene Blue (EMB) (**C**,**D**), and Mac Conkey agar (MCCA) (**E**,**F**) media for colony counts. IIRI: intestinal ischemia/reperfusion injury, FEB: febuxostat, LDPA: low-dose *Phyllanthus amarus*, HDPA: high-dose *Phyllanthus amarus*, ^a^ *p* < 0.05 versus sham, ^b^ *p* < 0.05 versus IIRI, ^c^ *p* < 0.05 versus FEB + IIRI. Values represent the mean for ten replicates ± the standard deviation.

**Figure 11 molecules-27-05073-f011:**
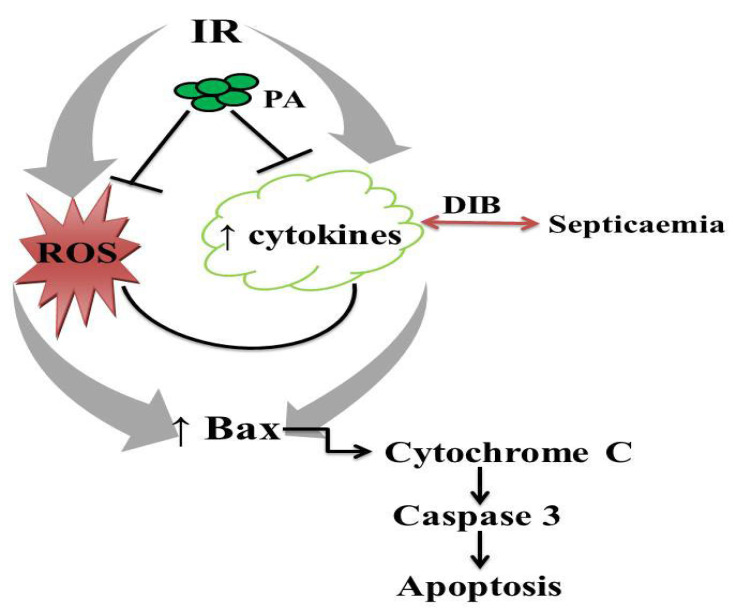
Schematic illustration of the effect of ischemia/reperfusion (IR) and methanolic *Phyllanthus amarus* leaf extract (PA) on Bax/caspase 3 signaling.

**Table 1 molecules-27-05073-t001:** Phytochemical constituents of methanolic *P. amarus* leaf extract.

Phytochemicals	Values
Saponins	0.747 ± 0.017
Tannins	1.850 ± 0.030
Phenolics	1.130 ± 0.014
Anthocyanin	1.820 ± 0.020
Alkaloids	1.607 ± 0.013
Triterpenoids	0.140 ± 0.010
Glycosides	0.863 ± 0.015

Values represent the mean for three replicates ± standard deviation.

**Table 2 molecules-27-05073-t002:** Organic constituents of methanolic *P. amarus* leaf extract revealed by GC-MS.

S/N	Compound	Molecular Weight (g/mol)	Molecular Formula	Area (%)	Retention Time	Quality
**1**	Butanoic acid	88.11	CH_4_H_8_O_2_	10.75	3.574	9
**2**	Methenamine	140.186	C_6_H_12_N_4_	7.58	3.797	5
**3**	Cyclobutane	56.107	C_4_H_8_	0.04	7.476	9
**4**	Octanoic acid	144.211	C_8_H_16_O_2_	0.17	9.828	89
**5**	Thiazole	85.12	C_3_H_3_NS	0.43	10.746	28
**6**	Oleic acid	282.468	C_18_H_24_O_2_	0.14	11.357	30
**7**	Citric acid	192.124	C_6_H_8_O_7_	1.53	12.129	25
**8**	Acetic acid	128.942	C_2_H_2_CL_2_O_2_	3.04	14.578	9
**9**	Hexadeconoic acid, methyl ester	270.45	C_17_H_34_O_2_	6.00	15.094	98
**10**	Methyl stearate	298.5	C_19_H_38_O_2_	2.41	16.655	95

**Table 3 molecules-27-05073-t003:** Effect of intestinal ischemia/reperfusion (I/R) and methanolic *Phyllanthus amarus* leaf extract on hepatic injury markers.

	Sham	IIRI	FEB + IIRI	LDPA + IIRI	HDPA + IIRI
AST (U/g tissue)	35.40 ± 4.27	84.80 ± 4.55 ^a^	46.40 ± 1.52 ^a,b^	40.51 ± 4.89 ^b^	37.40 ± 1.67 ^b,c^
ALT (U/g tissue)	54.67 ± 9.86	87.67 ± 9.29 ^a^	58.33 ± 2.88 ^b^	47.67 ± 4.50 ^b^	56.33 ± 3.51 ^b^
ALP (U/g tissue)	50.80 ± 2.58	86.34 ± 8.33 ^a^	59.60 ± 7.89 ^b^	57.52 ± 8.42 ^b^	54.01 ± 4.06 ^b^
GGT (U/g tissue)	123.51 ± 5.86	178.30 ± 7.70 ^a^	142.12 ± 4.40 ^a,b^	147.35 ± 7.59 ^a,b^	137.41 ± 9.10 ^b^

IIRI: intestinal ischemia/reperfusion injury, FEB: febuxostat, LDPA: low-dose *Phyllanthus amarus*, HDPA: high-dose *Phyllanthus amarus*, ^a^
*p* < 0.05 versus sham, ^b^ *p* < 0.05 versus IIRI, ^c^ *p* < 0.05 versus FEB + IIRI. Values represent the mean for ten replicates ± the standard deviation.

**Table 4 molecules-27-05073-t004:** Effect of intestinal ischemia/reperfusion (I/R) and methanolic *Phyllanthus amarus* leaf extract on oxidative stress markers and antioxidants.

	Sham	IIRI	FEB + IIRI	LDPA + IIRI	HDPA + IIRI
MDA (mM)					
Intestinal	0.683 ± 0.056	1.257 ± 0.051 ^a^	0.790 ± 0.036 ^b^	0.513 ± 0.032 ^a,b,c^	0.510 ± 0.037 ^a,b,c^
Hepatic	1.967 ± 0.305	5.567 ± 0.473 ^a^	1.967 ± 0.513 ^b^	2.400 ± 0.4359 ^b^	2.167 ± 0.305 ^b^
GSH (nmoL/g tissue)					
Intestinal	57.470 ± 3.522	32.430 ± 3.017 ^a^	58.800 ± 3.297 ^b^	78.600 ± 3.396 ^a,b,c^	78.570 ± 3.247 ^a,b,c^
Hepatic	63.130 ± 2.259	33.100 ± 3.751 ^a^	63.470 ± 2.721 ^b^	83.600 ± 3.396 ^a,b,c^	83.230 ± 2.673 ^a,b,c^
SOD (U/g)					
Intestinal	17.330 ± 0.493	12.500 ± 0.458 ^a^	24.470 ± 0.566 ^a,b^	16.130 ± 0.611 ^b,c^	18.130 ± 0.611 ^b,c,d^
Hepatic	8.333 ± 0.251	3.900 ± 0.435 ^a^	7.533 ± 0.404 ^b^	7.433 ± 0.321 ^b^	8.167 ± 0.305 ^b^
Catalase (mU/mg)					
Intestinal	37.000 ± 2.400	23.000 ± 2.000 ^a^	36.000 ± 2.210 ^b^	34.330 ± 2.517 ^b^	43.670 ± 2.517 ^a,b,c,d^
Hepatic	33.000 ± 2.000	20.000 ± 2.646 ^a^	29.670 ± 2.082 ^b^	30.000 ± 2.646 ^b^	32.670 ± 2.082 ^b^
GPx (nm/min/mg protein)					
Intestinal	161.900 ± 11.360	92.310 ± 5.473 ^a^	153.900 ± 12.590 ^b^	144.800 ± 7.333 ^b^	144.800 ± 8.951 ^b^
Hepatic	185.600 ± 19.240	108.900 ± 7.001 ^a^	158.400 ± 8.445 ^a,b^	148.300 ± 11.960 ^a,b^	157.000 ± 9.123 ^a,b^
Thiol protein (U/mg protein)					
Intestinal	16.430 ± 0.503	7.500 ± 0.360 ^a^	15.500 ± 0.400 ^b^	16.400 ± 0.625 ^b^	15.570 ± 0.416 ^b^
Hepatic	18.430 ± 0.503	6.500 ± 0.361 ^a^	17.830 ± 0.945 ^b^	18.400 ± 0.625 ^b^	17.800 ± 0.360 ^b^
Non-thiol protein (U/mg protein)					
Intestinal	28.370 ± 1.097	18.030 ± 0.650 ^a^	32.230 ± 0.776 ^a,b^	22.070 ± 0.802 ^a,b,c^	24.240 ± 0.686 ^a,b,c^
Hepatic	30.700 ± 1.082	18.370 ± 1.350 ^a^	34.230 ± 0.776 ^a,b^	25.070 ± 0.802 ^a,b,c^	27.240 ± 0.686 ^a,b,c^

IIRI: intestinal ischemia/reperfusion injury, FEB: febuxostat, LDPA: low-dose *Phyllanthus amarus*, HDPA: high-dose *Phyllanthus amarus*, MDA: malondialdehyde, GSH: reduced glutathione, SOD: superoxide dismutase, GPx: glutathione peroxidase, ^a^
*p* < 0.05 versus sham, ^b^ *p* < 0.05 versus IIRI, ^c^ *p* < 0.05 versus FEB + IIRI, ^d^ *p* < 0.05 versus LDPA + IIRI. Values represent the mean for ten replicates ± the standard deviation.

**Table 5 molecules-27-05073-t005:** Pearson’s correlation between hepatic injury markers and intestinal histomorphological distortion.

		Eckhoff’s Score	AST	ALT	GGT
Chiu’s score	Pearson correlation	0.8551	0.933	0.8822	0.2406
	R^2^	0.7312	0.8704	0.7784	0.0578
	Sig. (2-tailed)	<0.0001 *	<0.001 *	<0.001 *	0.388
	N	50	50	50	50

* *p* < 0.05.

## Data Availability

The data that support the findings of this study are available on request from the corresponding author. The data are not publicly available due to privacy or ethical restrictions.

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
