# Peer review of "Restoration of Hepatic and Intestinal Integrity by Phyllanthus amarus Is Dependent on Bax/Caspase 3 Modulation in Intestinal Ischemia-/Reperfusion-Induced Injury"

_molecules, 2022, doi:10.3390/molecules27165073_

Round 1
Reviewer 1 Report
This is a reasonably well conducted study that showcases the interest of plant extracts for the treatment of IRII, a phenomenon of wide interest in medicine.
I propose the following alterations for proceeding further with the manuscript:
AST are not liver specific. The authors should evaluate the extent of hepatic injury using liver specific ensymatic tests such as ALT activity tests.
Figure 4 data are distorted and the shape of the figure should be improved before resubmission of the revised version.
Figure 5: How many animals have been considered for the staining experiments? The authors should find a way to quantifiy the data, taking for example into account the density of vili and of leucocytic infiltrates in each sample category, together with statistics.
Figure 6: Likewise for the liver histomorphology that was considered: for instance, what is the quantified length and width of sinusoids that can be considered across all animals, together with statistical tests.
Figure 9 data should be monitored using an immunoblot approach followed by quantification.
Author Response
Comments and Suggestions for Authors
This is a reasonably well conducted study that showcases the interest of plant extracts for the treatment of IRII, a phenomenon of wide interest in medicine.
Response: Thanks.
I propose the following alterations for proceeding further with the manuscript:
AST are not liver specific. The authors should evaluate the extent of hepatic injury using liver specific ensymatic tests such as ALT activity tests.
Response: Thanks. Data on ALT activity have been included.
Figure 4 data are distorted and the shape of the figure should be improved before resubmission of the revised version.
Response: Thanks. Figure 4 has been improved.
Figure 5: How many animals have been considered for the staining experiments? The authors should find a way to quantifiy the data, taking for example into account the density of vili and of leucocytic infiltrates in each sample category, together with statistics.
Response: Thanks. The number of replicates has been stated. Also, data of the quantification of the villi length and crypt depth has been included.
Figure 6: Likewise for the liver histomorphology that was considered: for instance, what is the quantified length and width of sinusoids that can be considered across all animals, together with statistical tests.
Response: Thanks. Data of the quantification of the hepatic sinusoidal length and width has been included.
Figure 9 data should be monitored using an immunoblot approach followed by quantification.
Response: Thanks. Although data on immunoblot would enhance the present findings; since ELISA has been used for protein quantification, protein quantification using immunoblot may not add more value. Rather histopathological examination and immunohistochemistry were employed to confirm the mechanisms explored.
Reviewer 2 Report
In this study, the authors tried to investigate the role of Bax/caspase 3 signaling in intestinal ischaemia/reperfusion (I/R) induced intestinal and hepatic injury, while testing methanolic Phyllanthus Amarus leaf extract.
The study is well designed and properly executed, but I have the following major concern which needs to be addressed.
1. There is lack of detailed phytochemical analysis as the authors only mentioned that the extract is rich in tannins after GCMS analysis.
2. How the authors identified the detail constituents of methanolic extract on GCMS, please mention? It looks hard to characterized polar extract of plants through GCMS.
3. The extract also contains alkaloids as mentioned in page 5, so the action of extract may be due to the presence of alkaloids, so how the author assumed the tannins and anthocyanins.
4. the authors need to support the mechanism-based potential of the extract through chemical compounds rather than to assume a class of secondary metabolite.
5. All the markers were identified through ELISA kits, will be nice to perform western blots for more confirmation of the mechanism.
6. The authors mentioned Caspase-3, is its cleaved caspase -3? please mention.
7. There are also some minor mistakes which needs to be rectified.
Author Response
Comments and Suggestions for Authors
In this study, the authors tried to investigate the role of Bax/caspase 3 signaling in intestinal ischaemia/reperfusion (I/R) induced intestinal and hepatic injury, while testing methanolic Phyllanthus Amarus leaf extract.
The study is well designed and properly executed, but I have the following major concern which needs to be addressed.
Response: Thanks.
- There is lack of detailed phytochemical analysis as the authors only mentioned that the extract is rich in tannins after GCMS analysis.
Response: Thanks. Detailed phytochemical findings are included. The extract was found to be rich not only in tannins, but also in anthocyanin, alkaloids, and phenolics (Table 1). The findings of GCMS are in table 2.
- How the authors identified the detail constituents of methanolic extract on GCMS, please mention? It looks hard to characterized polar extract of plants through GCMS.
Response: Thanks. Details of the GCMS are included in section 2.4.
- The extract also contains alkaloids as mentioned in page 5, so the action of extract may be due to the presence of alkaloids, so how the author assumed the tannins and anthocyanins.
Response: Thanks. Authors did not assume that tannins and anthocyanins are responsible for the actions of the phytochemicals. It has been clearly stated in the abstract and body of the manuscript that in tannins, anthocyanin, alkaloids, and phenolics may be responsible. The likely roles of tannins, anthocyanin, alkaloids, and phenolics were also briefly discussed in relation to the extract.
- the authors need to support the mechanism-based potential of the extract through chemical compounds rather than to assume a class of secondary metabolite.
Response: Thanks. The mechanism of action of the extract was explained using the constituent phytochemicals and molecules found on GCMS analysis.
- All the markers were identified through ELISA kits, will be nice to perform western blots for more confirmation of the mechanism
Response: Thanks. Although most markers were identified through ELISA kits, some were identified through histopathological evaluation and immunohistochemistry.
- The authors mentioned Caspase-3, is its cleaved caspase -3? please mention.
Response: Thanks. Caspase 3 activity was assayed. This has been clarified in text and figure.
- There are also some minor mistakes which needs to be rectified.
Response: Thanks. All comments by reviewers have been thoroughly addressed.
Round 2
Reviewer 1 Report
The study has improved and is ready for publication. Improving the promotion of such approaches is of valuable interest in the field of chronic liver disease as well.
Reviewer 2 Report
response of all questions were found satisfactory.